# The Effect of Earthing Mat on Stress-Induced Anxiety-like Behavior and Neuroendocrine Changes in the Rat

**DOI:** 10.3390/biomedicines11010057

**Published:** 2022-12-26

**Authors:** Hyun-Jung Park, Woojin Jeong, Hyo Jeong Yu, Minsook Ye, Yunki Hong, Minji Kim, Ji Youn Kim, Insop Shim

**Affiliations:** 1Department of Food Science and Biotechnology, Kyonggi University, 154-42, Gwanggyosan-ro, Youngtong-gu, Suwon 17104, Gyeonggi-do, Republic of Korea; 2Department of Physiology, College of Medicine, Kyung Hee University, 26 Kyung Hee-daero, Seoul 02447, Republic of Korea; 3Department of East West Science, Graduate School of East-West Medical Science, Kyung Hee University, 1732 Deogyeong-dae ro, Giheung-gu, Yongin-si 17104, Gyeonggi-do, Republic of Korea; 4Graduate School of Global Pharmaceutical Industry and Clinical Pharmacy, Ajou University, 206 Worldcup-ro, Youngtong-gu, Seoul 02447, Republic of Korea

**Keywords:** grounding, earthing mat, stress, elevated plus maze (EPM), corticotrophin releasing factor (CRF), c-Fos, depressive behavior, tail suspension test (TST), forced swimming test (FST), therapeutic technique

## Abstract

Grounding is a therapeutic technique that involves doing activities that “ground” or electrically reconnect us to the earth. The physiological effects of grounding have been reported from a variety of perspectives such as sleep or pain. However, its anti-stress efficacy is relatively unknown. The present study investigated the stress-related behavioral effects of earthing mat and its neurohormonal mechanisms in the Sprague–Dawley male rat. Rats were randomly divided into four groups: the naïve normal (Normal), the 21 days immobilization stressed (Control), the 21 days stressed + earthing mat for 7 days (A7) or 21 days (A21) group. The depressive-and anxiety like behaviors were measured by forced swimming test (FST), tail suspension test (TST) and elevated plus maze (EPM). Using immunohistochemistry, the expression of corticotrophin-releasing factor (CRF) and c-Fos immunoreactivity were analyzed in the brain. In the EPM, time spent in the open arm of the earthing mat groups was significantly increased compared to the Control group (*p* < 0.001), even though there were without effects among groups in the FST and TST. The expression of CRF immunoreactive neurons in the earthing mat group was markedly decreased compared to the Control group. Overall, the earthing mat reduced stress-induced behavioral changes and expression of c-Fos and CRF immunoreactivity in the brain. These results suggest that the earthing mat may have the potential to improve stress-related responses via the regulation of the corticotrophinergic system.

## 1. Introduction

Repeated immobilization stress is an easy and well-known method to induce chronic physical and emotional stress [1]. The psychological and physiological changes to repeated immobilization stress are initiated by activation of the hypothalamic–pituitary adrenal axis, and these results in the release of catecholamines and stress hormones such as corticotropin-releasing factor (CRF) [2,3]. The CRF system plays a key role in a diversity of behaviors accompanying stress, anxiety and depression [4,5,6]. We previously demonstrated increased CRF expression in the paraventricular nuclei of repeated restraint-stressed rats [6]. However, little is known about the specific interventions for stress-related disorders.

Grounding, also called earthing, is a therapeutic technique that involves doing activities that “ground” or electrically reconnect us to the earth. This conductive contact of the human body with the surface of the earth can have intriguing benefits on our physiology. Recently, researchers have studied the physiological effects of grounding from a variety of perspectives [7,8]. For example, electrically conductive contact of the human body with the surface of the Earth (grounding or earthing) produces intriguing effects on physiology and health [9,10]. Chevalier et al. reported that grounding reduces pain and alters the numbers of circulating neutrophils and lymphocytes, and also affects various circulating chemical factors related to inflammation [9]. Some studies reported that environmental medicine generally influences environmental factors with a negative impact on human health [7,9]. Mounting evidence shows that the Earth’s negative potential can produce a stable internal bioelectrical environment for the normal functioning of all body systems [8,9,10]. It is known that electrons from antioxidant molecules normalize reactive oxygen species involved in immune, inflammatory and stress response. Therefore, it is possible that the influx of free electrons absorbed into the body through direct contact with the Earth normalize free radicals and may reduce stress vulnerability. However, no studies investigated the anti-stress effects or mechanism of earthing mat underlying stress responses. The main hypothesis of this study is that connecting the body to the earth through earthing mat may have anti-inflammatory and antioxidant effects and, therefore, exposure with earthing mat has an anti-stress efficacy in animal models of stress.

In this study, we aimed to investigate whether grounding in the rats could change stress-related anxiety and depressive behaviors and the production of corticotrophin-releasing factors in the brain region. To achieve this goal, anxiety and depressive-like behaviors were tested via an elevated plus maze (EPM), tail suspension test (TST) and forced swimming test (FST). Moreover, we further assessed the expression of c-Fos and corticotrophin-releasing factor (CRF) in the paraventricular nucleus of the hypothalamus (PVN).

## 2. Materials and Methods

### 2.1. Animals and Experimental Procedures

All the experiments were approved by the Kyung Hee University institutional animal care (approval No. KHUAP(SE)-13-041) and in accordance with the US National Institutes of Health “Guide for the Care and Use of Laboratory Animals” (NIH Publication number 80-23, revised 1996). Sprague–Dawley rats (Orient Animal Co, Kyonggido, Korea) that weighed 220–240 g (6–7 weeks old) each were used for the experiment. The male rats were housed under a controlled temperature (22–24 °C) with a 12 h light/dark cycle. The lights were on from 8:00 to 20:00. Food and water were made available ad libitum. The rats were allowed at least 1 week to adapt to their environment before the experiments. All the experiments were approved by the Kyung Hee University institutional animal care and use committee. The male rats were randomly divided into four groups: the naïve Normal (Normal), the 21 days stressed group (Control), the 21 days stressed and the earthing mat used the group for 7 days (A7) or 21 days (A21) (Figure 1).

Immobilization stress was applied by forcing the animals into an immobilization device (a disposable rodent restraint cone, Yusung, Korea) 2 h (10:00–12:00 a.m.) per day for 21 consecutive days. For the 21 days immobilization stressed group (Control), immobilization stress was applied by placing rats into a disposable rodent restraint cone 2 h per day for 21 consecutive days, and they were not placed in cage with earthing mattress during experiments. For 7 days (A7) or 21 days (A21) groups, they received immobilization stress for 21 consecutive days and were placed in cage with earthing mattress, as shown in Figure 2. Neither immobilization stress nor earthing mat was applied to the normal group (Normal). When the mats were destroyed by rat, they were replaced. Otherwise, the same one was used for one rat. They were cleaned and sanitized with 70% alcohol swab every other day during experiment.

### 2.2. Earthing Mat

The earthing mat was provided by the World Home Dr. Company (Anyang City, Kyunggido, Republic of Korea). The earthing mat system consists of cotton sheet, and electric emission plate is connected to a ground port of an electrical outlet, as shown in Figure 2. The grounding port helps reconnect the conductive rats’ bodies to the Earth’s natural and subtle surface electric charge. Rats were placed on the earthing mat before behavioral tests.

### 2.3. Tail Suspension Test (TST)

All mice were isolated in plastic boxes (20 × 10 × 10 cm) between the injection and the test. It is easy to measure the length of immobility, and these measurements were always made under blind conditions [11,12,13]. The recording duration was 6 min.

### 2.4. Forced Swimming Test (FST)

Transparent Plexiglas cylinders (height: 50 cm × diameter: 20 cm) were required for forced swimming test (FST). The room temperature water was filled to a 30 cm depth. Before test, all rats were taken pre-test for 15 min. After 24 h, rats were tested for 5 min. We used a video camera, and swimming behaviors were analyzed. Total duration of immobility (lack of motion of the whole body, climbing (vigorous movements) and swimming (large forepaw movements displaced the body around the cylinder) were examined [6,14,15,16].

### 2.5. Elevated Plus Maze (EPM)

The plus-maze apparatus was constructed with black wood. It consisted of two open arms (the arms extended from a central 50 × 10 cm space) and two enclosed arms (50 × 10 × 40 cm). The arms extended from a central platform (10 × 10 cm). The apparatus was elevated 50 cm above the floor. The animals were transported to the testing room at least 1 h prior to starting the experiment. The rats were individually placed in the central platform facing a closed arm, and they were allowed to explore the maze for a 5-min test period. The duration of time spent in the open and closed arms, and activity of each arm were the behavioral measures that were recorded for each rat [17,18,19,20,21,22]. The apparatus was wiped clean with a damp sponge and dried with paper towels between tests.

### 2.6. Immunohistochemistry of c-Fos and Corticotrophin Releasing Factor (CRF)

After the behavioral tests, rats were perfused through a needle in the left ventricle of the heart, under sodium pentobarbital (100 mg/kg, i.p.) with 100 mL of saline for 5 min, followed by approximately 500 mL of a 4% solution of formaldehyde in PBS. After perfusion, the brains were removed from skull and post-fixed in the same fixative solution for 2 h at 4 °C and then placed overnight at 4 °C in 20% sucrose in PBS. They were frozen and cut on microtome. The coronal sections were sliced to 30 μm-thickness. CRF [23,24,25,26,27,28] and c-Fos [29,30,31,32,33,34,35] immunohistochemistry were performed separately. Primary antibodies were diluted with blocking solution (rabbit CRF polyclonal antibody, concentration 1:500; Santacruz biotechnology, Delaware Avenue Santa Cruz, CA, USA.) and c-Fos immunoreactivity using rabbit c-Fos polyclonal antibody (c-Fos, concentration 1:2000; Santacruz biotechnology, Delaware Avenue Santa Cruz, CA, USA). Sections were incubated overnight 18 h–24 h, free-floating) at room temperature with gentle agitation. Following rinsing in PBS, the sections were incubated for 2 h at room temperature in biotinylated rabbit anti-rabbit serum (Vector Laboratories, Burlingame, CA, USA) that was diluted 100:1 in PBST containing 2% normal goat serum. The sections were placed in Vectastain Elite ABC reagent (Vector Laboratories, Burlingame, CA, USA) for 2 h at room temperature. Following a further rinsing in PBS, the tissue was developed using diaminobenzidine chromogen with nickel intensification. Sections were washed in 1× PBS three times for 3 min, mounted on gelatin-coated slides, air-dried (2 h) and cover-slipped with mounting solution. Microscopy was used to acquire digital images of CRF immunohistochemistry in paraventricular nucleus using a 100× objective from 3–4 tissue sections from each animal. Then, a micro rectangular grid (200 × 200 μm) was placed on PVN area (Bregma ML: 0~−0.8 mm, AP: −1.5~−2 mm, DV: −7.8~−8 mm) according to the atlas of Paxinos and Watson [36].

### 2.7. Statistical Analysis

The values of the experimental results were expressed as the mean ± S.E.M. Statistical analysis was used with SPSS 25.0 software (SPSS 25 Inc., Chicago, IL, USA). Differences among groups were analyzed using one-way ANOVA and LSD post hoc test. *p*-value of less than 0.05 was considered statistically significant. Graph generations were followed with GraphPad Prism 6.0 software.

## 3. Results

### 3.1. Forced Swimming Test (FST) and Tail Suspension Test (TST)

As shown in Figure 3A–C, in the FST, the immobility time was not significantly different among groups (immobility (F (3, 50) = 0.53, *p* = 0.67, Figure 3A), climbing (F (3, 50) = 0.08, *p* = 0.97, Figure 3B) and swimming (F (3, 50) = 0.07, *p* = 0.98, Figure 3C). In the FST and TST, the immobility time of the earthing mat groups (A7 and A21) tended to decrease more compared to the Control group. (Figure 3A,D).

### 3.2. Elevated Plus Maze

As shown in Figure 4A–C, the one-way ANOVA revealed a significant difference among groups regarding the time to the open arms (%) (F (3, 19) = 9.2, *p* = 0.0006, Figure 4A) and the closed arm (%) (F (3, 19) = 3.6, *p* = 0.03, Figure 4B). The control group decreased the time spent in the open arms (*p* < 0.05) compared to that of the normal group. Earthing mat-used groups showed a significantly longer period of time spent in the open arms than that of the control group (*p* < 0.001). The control group increased the time spent in the closed arms (*p* < 0.05) compared to that of the normal group. The group using the Earthing mat for seven days showed a significantly shorter period of time spent in the closed arms than that of the control group (*p* < 0.05). However, the total distance was not significantly different among groups (F (3, 19) = 1.3, *p* = 0.29, Figure 4C). This result showed that earthing mat-used groups produced an anxiolytic effect on the EPM.

### 3.3. Immunohistochemistry

#### 3.3.1. Corticotrophin-Releasing Factor (CRF) Immunohistochemistry

The evaluation of the CRF immunoreactive cells per section of the paraventricular area is shown in Figure 5A,B. The number of CRF positive neurons in the paraventricular nucleus (PVN) area was 53.3.0 ± 6.2 in the Normal group, 89.9.8 ± 6.7 in the Control group, 63.5 ± 7.1 in the 7-day earthing mat group, 70.0 ± 4.6 in the 21-day earthing mat group [F (3, 188) = 6.1, *p* = 0.0005]. When resulted in the expression of the CRF, CRF immunoreactive neurons in the control group were significantly increased compared to the Normal group. However, the expression of CRF immunoreactive neurons in the earthing mat groups was markedly decreased compared to the Control group.

#### 3.3.2. c-Fos Immunohistochemistry

The evaluation of the c-Fos immunoreactive cells per section of the paraventricular area is shown in Figure 6A,B. The number of c-Fos positive neurons in the paraventricular nucleus (PVN) area was 45.3 ± 4.6 in the Normal group, 42.4 ± 4.4 in the Control group, 39.6 ± 2.7 in the 7-day earthing mat group, 46.9.0 ± 3.3 in the 21-day earthing mat group, [F (3, 187) = 0.7, *p* = 0.54]. The expression of the c-Fos immunoreactive neurons in the earthing mat seven-day group had a trend of more decreased c-Fos neurons compared to the control group. However, there were no significant differences among the groups.

## 4. Discussion

The present study showed that in the elevated plus maze (EPM), the Control group decreased spent time in the open arm compared to the Normal group. However, earthing mat groups significantly increased spent time in the open arm compared to the Control group. When resulted in the expression of the CRF, CRF immunoreactive neurons in the Control group were significantly increased compared to the Normal group. However, the expression of CRF immunoreactive neurons in the earthing mat groups was markedly decreased compared to the Control group. The expression of the c-Fos immunoreactive neurons in the seven-day earthing mat group trended to decrease more than in the Control group. However, there were no significant differences among the groups.

Immobilization stress is one of the main and potent sources of stress, inducing a strong hormonal and behavioral reaction [37]. The HPA axis is activated in rodents to a different degree when the stress is mild (e.g., mild handling, needle stick, time in elevated plus maze [38], common symptoms of stress-related behavior [39,40]. A decrease in general exploratory activity in an open arena after restraint stress has been previously described [41]. In the elevated plus-maze, most studies found a decrease in the percentage of open-arm entries and/or time spent in them [41,42]. Consistent with the previous study, the present result showed a decrease in time spent in the open arm after repeated stress. However, an increase in time spent in the open arms was shown in earthing mat groups. Our result also showed that the earthing mat decreased anxiety-like behavior compared to the Control group. Another pilot study showed that grounding improves measurements of mood within 1 h, suggesting a potentially positive effect on health [43,44,45,46,47,48]. Therefore, it can be assumed that earthing mat may be effective against stress-related anxiety.

It is well known that repeated stress has an effect on the central nervous system (CNS)-neuroendocrine behavior. The immunohistochemical expression of c-Fos-like proteins in the nervous system is considered a marker of neuronal activation. The logic of this methodology is based on the demonstration that the expression of this protein is increased in neurons after stress exposure [49,50]. The expression of the c-Fos immunoreactive neurons in the seven-day earthing mat group trended to decrease more than in the Control group. However, there were no significant differences among the groups.

Corticotropin-releasing factor (CRF) is a key component of stress responsivity, modulating related behaviors, including anxiety and reward. The primary stress response involves the activation of hypothalamic neurons producing CRF, an initial step in the cascade that leads to the synthesis and release of glucocorticoids [51,52,53,54,55,56,57]. Our data also showed the activation of CRF immunoreactive neurons in the PVN after repeated 21 days of restraint stress. However, earthing mat groups decreased the CRF-ir expression in the PVN compared to the Control group. Mounting evidence suggests that the Earth’s negative potential can create a stable internal bioelectrical environment for the normal functioning of all body systems [43,44,45,46]. Moreover, oscillations of the intensity of the Earth’s potential may be important for setting the biological clocks regulating diurnal body rhythms, such as cortisol secretion [51]. The results of the study were consistent with previous studies [18,51].

Earthing means reconnecting the conductive body to the Earth’s natural electric charge. Earthing influences the basic bioelectrical function of the body. The present study showed that earthing influenced corticosterone secretion and stress-related behavioral changes such as anxiety and learned helplessness. These results suggest that connecting the whole body to the earth can free electrons and diurnal electrical rhythms to enter the body, setting the biological clocks for hormones that regulate stress responses, consistent with previous studies proving that Earthing stabilizes the physiology, reduces inflammation and pain and improves sleep [43,44,45,46,47,58,59]. Another study reported that through the electrodynamics of red blood cells earthing significantly reduced blood viscosity and cardiovascular disease [60].

Taken together, these data suggest that earthing mats may be helpful in stress management via the regulation of corticotrophinergic mechanisms. In light of such limitations, our finding provides preliminary evidence of the safety of earthing mats and their potential to decrease stress responses in an animal model. Moreover, we are planning to analyze stress-related neurotransmitter markers such as serotonin, dopamine and GABA synthesis and release in the future study.

## 5. Conclusions

In summary, earthing mat reduced anxiety-like behavior and learned helplessness behavior. These behavioral alterations may be mediated via the regulation of the corticotrophinergic pathway. Overall, connecting the whole body to the earth can free electrons and diurnal electrical rhythms to enter the body, setting the biological clocks for hormones that regulate stress responses.

## Figures and Tables

**Figure 1 biomedicines-11-00057-f001:**
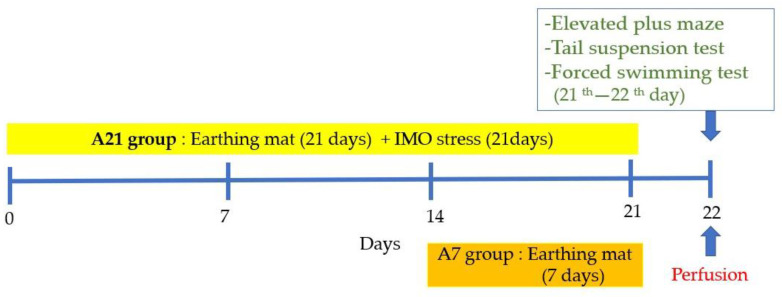
Experimental schedule.

**Figure 2 biomedicines-11-00057-f002:**
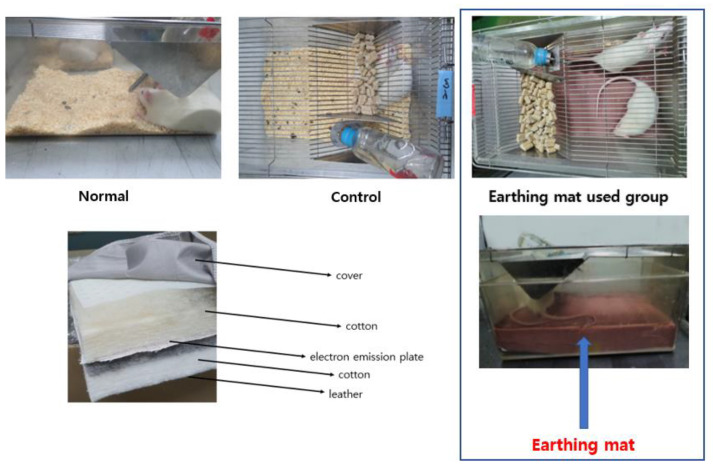
Composition of Earthing mat and grounded system.

**Figure 3 biomedicines-11-00057-f003:**
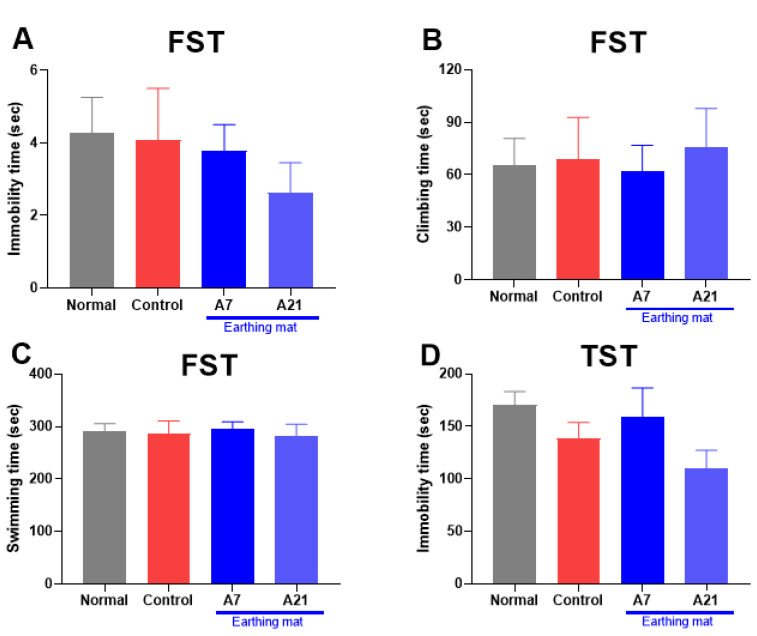
Effect of earthing mat on the FST (**A**–**C**) and TST (**D**). Data represent means ± SEM.

**Figure 4 biomedicines-11-00057-f004:**
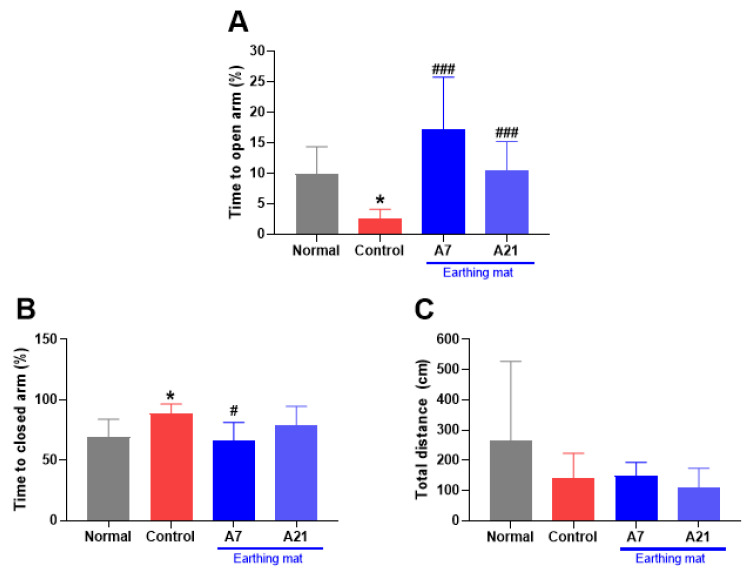
Effect of earthing mat on elevated plus maze. (**A**): Time to open arm (%), (**B)**: Time to closed arm (%), (**C**): Total distance (cm).

**Figure 5 biomedicines-11-00057-f005:**
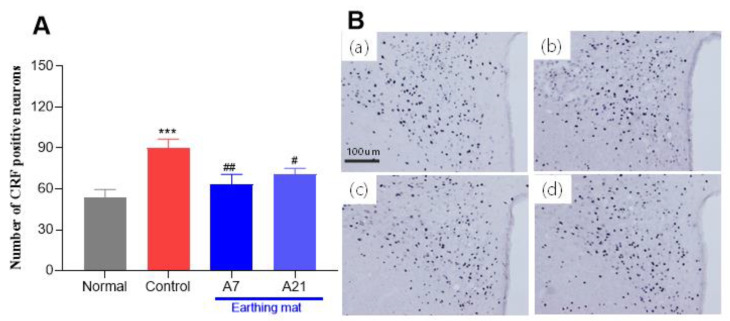
(**A**) Effect of earthing mat on CRF expression. (**A**) Data represent means ± SEM. ***** < 0.001 compared to Normal group, # *p* < 0.05, ## *p* < 0.01 compared to Control group. (**B**) Photographs showing the distribution of CRF immunoreactive cells in the PVN of the (**a**) Normal group, (**b**) Control group, (**c**) A7 group, (**d**) A21 group. Coronal sections were 30 μm thick, and the scale bar represents 100 μm.

**Figure 6 biomedicines-11-00057-f006:**
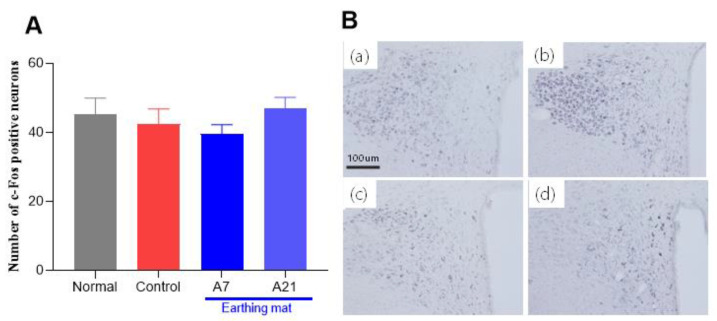
(**A**) Effect of earthing mat on c-Fos expression, Data represent means ± SEM. (**B**) Photographs showing the distribution of CRF immunoreactive cells in the PVN of the (**a**) Normal group, (**b**) Control group, (**c**) A7 group, (**d**) A21 group. Coronal sections were 30 μm thick, and the scale bar represents 100 μm.

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
