# Peer review of "The Effect of Earthing Mat on Stress-Induced Anxiety-like Behavior and Neuroendocrine Changes in the Rat"

_biomedicines, 2022, doi:10.3390/biomedicines11010057_

Round 1

Reviewer 1 Report

Dear authors.

The present manuscript is indeed interesting, abording the potential neuroprotective effects of earthing mat in stress related responses via regulation of corticotrophinergic system. However, the present version of the manuscript has several problems.

In general I think that the paper is very simplistic.

The text has several errors in text formation and also grammatical errors.

Globally, the methods should be much more detailed, it is not described with sufficient detail to allow others to replicate the study. For example, it is not described how earthing mat was performed. Is not mentioned the number of animals per group, etc. Another example, the authors write that used five groups but then there are only four groups in the graphs.

In the figures legends there are no reference to the immunohistochemistry photographs, so we do not know which group a specific photograph represent.

The discussion is simplistic.

Author Response

Response to reviewers of “Effect of earthing mat on stress relief in the rats” by Insop Shim 24th of Nov of 2022

biomedicines-1967572

We would like to thank the reviewers for their thoughtful review of the manuscript. They raise important issues, and their inputs are very helpful for improving the manuscript. We agree with almost all their comments, and we have revised our manuscript accordingly.

We are already crafting a revised version of the paper that it states the hypothesis and the implications of our work more clearly than before. Moreover, we are including all reviewers’ suggestions and clarifying the text when needed. We are confident that the new version of the manuscript will be greatly improved.

We respond below in detail to each of the reviewer’s comments. In addition, we include how we have revised things, or if we have slightly disagreed with something, we stated why. We hope that the reviewers will find our responses to their comments satisfactory, and we are willing to finish the revised version of the manuscript including any further suggestion that the reviewers may have.

Looking forward hearing from you soon.

Sincerely,

Insop shim

Author's Reply to the Review Report (Reviewer 1)

The present manuscript is indeed interesting, abording the potential neuroprotective effects of earthing mat in stress related responses via regulation of corticotrophinergic system. However, the present version of the manuscript has several problems. In general I think that the paper is very simplistic. The text has several errors in text formation and also grammatical errors.

Response: As reviewer’s comment, the revised version is more elaborated with details and grammatical errors were corrected.

Globally, the methods should be much more detailed, it is not described with sufficient detail to allow others to replicate the study. For example, it is not described how earthing mat was performed. Is not mentioned the number of animals per group, etc. Another example, the authors write that used five groups but then there are only four groups in the graphs.

Response: As reviewer’s comment, we have provided more detailed information on experimental procedures and design in Methods part. And we also have corrected edits of group description in revised one (See also, Lines 79--90).

In the figures legends there are no reference to the immunohistochemistry photographs, so we do not know which group a specific photograph represent.

Response: As reviewer’s comment, we improved the Figure 5 and Figure 6. (See also, 195-199 and 210-213)

The discussion is simplistic.

Response: As reviewer’s comment, we discussed it in deep and more details. (Lines 257-265)

Reviewer 2 Report

Comments to the Author

Comments:
Manuscript ID: biomedicines-1967572, Title: Effect of earthing mat on stress relief in the rats

General comments

The study presents original information on the effects of earthing mat on stress in rats. It is a short communication. Although the work is original and interesting, it needs future revisions, including a more concrete hypothesis regarding the possible effects of earthing mat, much more information on materials and methods is missing, as well as the discussion can be improved. It is suggested that the authors review a series of specific comments, which I hope will be helpful to the authors:

Specific comments

Abrstact

L18-20: Stress from immobilization is mentioned here, but in the M&M manuscript text it is not clarified which stressor, nor the time, nor the conditions, nor the moment of the day in which they were stressed, etc... Could the authors clarify and unify the information throughout the manuscript?

L19: In the abstract 4 groups are mentioned, but in Materials and methods of the manuscript 5 groups are mentioned, unify and clarify well.

L19: What do the authors mean by the group "the naïve normal (Normal)"? In that group, are the rats without the mat or without stress, or both?

L19-20: The group the 14 days immobilization stressed group (Control), did not have a mat? If it did not have a mat, what was the condition of the cage?

L23-24: I suggest taking out the word "significantly" and would put the p value in parentheses.

Keywords

Avoid repeating words that are already in the title.

Introduction

L38-39: “As Aristotle once said, “In 38 nature, there is something marvelous.” Do not the authors think that this phrase is too broad in relation to the "nature" comment? What is the real contribution of including this statement?

L49: Why "relatively"? Can you be more precise with the statement?

Could the authors include a hypothesis, and its corresponding justification?

Materials and methods

L68: Five groups are mentioned, but only 4 are described, are there 5 or 4 groups?

L68: How was the group of 14 days stressed? with what stressor? How often? What time of day?, etc. Please include all necessary information.

L68: The control and normal groups did not have a mat?

There was no difference in the body weight of the rats before and after the experiment?

How many animals were kept per box? What was the degree of kinship between them?

Figure 1:

The tail suspension test needs to be included. When was it done?

It is suggested to include in this figure the different groups used.

L76-77: Much more information is needed, what is the composition?, materials?, and other characteristics... Were the same ones always used or were they changed? If they were changed how often? If they were not changed, how were they cleaned and sanitized? Waiting times, and characteristics of the components... Please, it is suggested to clarify all the necessary details.

Figure 2:

If the control group and the normal group did not have a mat, what was the support of their cages like? Could you show photos of those groups?

L82-105: What was the sequence of tests used?

How much time elapsed between each test for each animal?

L108: How was the region of the paraventricular nucleus located and identified?

L133-138: All data had normal distribution?

How was the distribution of the variables analyzed?

Results

L148, 149, 150, etc.: It is suggested to put the exact p value.

In figure 4 numeral symbols appear without clarifying in the legend of the figure.

In Figure 5, four images appear that are not explained or mentioned in the legend of the figure.

Discussion

The discussion is poor, it is necessary to try to comment on what could be the reason for the effects of the mat? What would be the possible mechanisms of action?

L200: “study” or “studies”?

L220-221: Did the authors assess or measure the bioelectric potential of the mat used?

Author Response

Response to reviewers of “Effect of earthing mat on stress relief in the rats” by Insop Shim 24th of Nov of 2022

biomedicines-1967572

We would like to thank the reviewers for their thoughtful review of the manuscript. They raise important issues, and their inputs are very helpful for improving the manuscript. We agree with almost all their comments, and we have revised our manuscript accordingly.

We are already crafting a revised version of the paper that it states the hypothesis and the implications of our work more clearly than before. Moreover, we are including all reviewers’ suggestions and clarifying the text when needed. We are confident that the new version of the manuscript will be greatly improved.

We respond below in detail to each of the reviewer’s comments. In addition, we include how we have revised things, or if we have slightly disagreed with something, we stated why. We hope that the reviewers will find our responses to their comments satisfactory, and we are willing to finish the revised version of the manuscript including any further suggestion that the reviewers may have.

Looking forward hearing from you soon.

Sincerely,

Insop shim

Author's Reply to the Review Report (Reviewer 2)

Comments:
Manuscript ID: biomedicines-1967572, Title: Effect of earthing mat on stress relief in the rats

General comments

The study presents original information on the effects of earthing mat on stress in rats. It is a short communication. Although the work is original and interesting, it needs future revisions, including a more concrete hypothesis regarding the possible effects of earthing mat, much more information on materials and methods is missing, as well as the discussion can be improved. It is suggested that the authors review a series of specific comments, which I hope will be helpful to the authors:

Specific comments

Abrstact

L18-20: Stress from immobilization is mentioned here, but in the M&M manuscript text it is not clarified which stressor, nor the time, nor the conditions, nor the moment of the day in which they were stressed, etc... Could the authors clarify and unify the information throughout the manuscript?

Response: As Reviewer suggested, we clearly state procedure for stress and subject information in revised version as follows; The Sprague-Dawley male rats were randomly divided into four groups: the naïve normal (Normal), the 21 days immobilization stressed group (Control), the 21 days immobilization stressed + earthing mat for 7 days (A7) or 21days (A21) (See also, Lines 80-90)

L19: In the abstract 4 groups are mentioned, but in Materials and methods of the manuscript 5 groups are mentioned, unify and clarify well.

Response: We have corrected group numbers. (See also, Lines 77-79)

L19: What do the authors mean by the group "the naïve normal (Normal)"? In that group, are the rats without the mat or without stress, or both?

 Response: The Normal group rats were defined as the group which was neither placed in the earthing mat nor underwent stress. (See also, Lines 78-79).

L19-20: The group the 14 days immobilization stressed group (Control), did not have a mat? If it did not have a mat, what was the condition of the cage?

Response: For the 21 days immobilization stressed group (Control), immobilization stress was applied by placing rats into a disposable rodent restraint cone 2 h per day for consecutive 21 days, and they were not placed in cage with earthing mattress during experiments. Neither immobilization stress nor earthing mat were applied to the normal group.   

(See also, Lines 77-79)

L23-24: I suggest taking out the word "significantly" and would put the p value in parentheses.

Response: In the EPM, time spent in the open arm in the earthing mat groups was significantly increased compared to the Control group (P < 0.001).  (See also, Line 29)

Keywords

Avoid repeating words that are already in the title.

Response: We delete “’earthing mat and stress” in the key words.

Introduction

L38-39: “As Aristotle once said, “In 38 nature, there is something marvelous.” Do not the authors think that this phrase is too broad in relation to the "nature" comment? What is the real contribution of including this statement?

Response: We delete this sentence “’As Aristotle~ marvelous”. (See also, 38-39).

L49: Why "relatively"? Can you be more precise with the statement?

Response: We delete “relatively”

Could the authors include a hypothesis, and its corresponding justification?

It is well known that electrons from antioxidant molecules normalize reactive oxygen species involved in immune, inflammatory and stress response. Therefore, it is possible that the influx of free electrons absorbed into the body through direct contact with the Earth normalize free radicals and may reduce stress vulnerability. However, no studies investigated the anti-stress effects or mechanism of earthing mat for stress responses.

The main hypothesis of this study is that connecting the body to the earth through earthing mat may have anti-inflammatory and antioxidant effects and therefore exposure with earthing mat has an anti-stress efficacy in animal models of stress. We have tested and justified this hypothesis and idea in the present study. We added this sentence in the revised one. (See also, Lines 52-59)

Materials and methods

L68: Five groups are mentioned, but only 4 are described, are there 5 or 4 groups?

Response: We have corrected this error, 4 groups. (See also, Lines 77-79)

L68: How was the group of 14 days stressed? with what stressor? How often? What time of day?, etc. Please include all necessary information.

Response: As reviewer’s comment, we included all information in methods of the revised one. (See also, Lines 78-79)

L68: The control and normal groups did not have a mat?

Response: The normal and control groups do not have a mat.

There was no difference in the body weight of the rats before and after the experiment?

Response: As reviewer’s comment, we examined the body weight during the experiment, but we did not observe any significant differences among the groups.

How many animals were kept per box? What was the degree of kinship between them?

Response: Two animals were kept per 1 box.

Figure 1:

The tail suspension test needs to be included. When was it done?

It is suggested to include in this figure the different groups used.

Response: As reviewer’s comment, we included the TST in revised manuscript as Figure 3D.

L76-77: Much more information is needed, what is the composition?, materials?, and other characteristics... Were the same ones always used or were they changed? If they were changed how often? If they were not changed, how were they cleaned and sanitized? Waiting times, and characteristics of the components... Please, it is suggested to clarify all the necessary details.

Response: As reviewer’s comment, we clearly state the composition of Earthing mat and grounded system in Figure 2 and text in details in Methods. If the mats were destroyed by rat, and then they were replaced. Otherwise, the same one was used for one rat. They were cleaned and sanitized with 70% alcohol swab every other days during experiment (see also, Lines 87-90 and Figure 2.)

Composition of Earthing mat and grounded system in Figure 2:

If the control group and the normal group did not have a mat, what was the support of their cages like? Could you show photos of those groups?

Response: As reviewer’s comment, we have included the representative photos of three groups.

L82-105: What was the sequence of tests used?

How much time elapsed between each test for each animal?

Response: As reviewer’s comment, we add the experimental schedule in the Figure 2. Also, 30 min elapsed between each test for each animal.

L108: How was the region of the paraventricular nucleus located and identified?

Response: Brain regions analyzed for CRF and c-Fos expression are based on the Paxinos and Watson brain atlas (Bregma ML: 0 ~ -0.8 mm, AP: -1.5 ~ -2 mm, DV: -7.8 ~ -8 mm) (See also, Lines 152-153)

L133-138: All data had normal distribution?

How was the distribution of the variables analyzed?

Response: All experimental data showed normal distribution. The values of the experimental results were expressed as the mean ± S.E.M. Statistical analysis was used with SPSS 25.0 software (SPSS 25 Inc., Chicago, IL). Differences among groups were analyzed using one-way ANOVA and LSD post hoc test. P-value of less than 0.05 was considered statistically significant. Graph generations were followed with GraphPad Prism 6.0 software. (See also, Lines  156-160)

Results

L148, 149, 150, etc.: It is suggested to put the exact p value.

Response: As reviewer’s comment, we put the exact p value. (See also, Line 164-165, Line 172-173, Line 180, Line 191, Line 207)

In figure 4 numeral symbols appear without clarifying in the legend of the figure.

Response: As reviewer’s comment, we revised numeral symbols in the Figure 4.

In Figure 5, four images appear that are not explained or mentioned in the legend of the figure.

Response: As reviewer’s comment, we improved the Figure 5 and Figure 6. (See also, 195-199 and 210-213)

Discussion

The discussion is poor, it is necessary to try to comment on what could be the reason for the effects of the mat? What would be the possible mechanisms of action?

Response: As reviewer’s comment, we discuss our data in deep. (Lines 257-266)

L200: “study” or “studies”?

Response: As reviewer’s comment, we have corrected several minor errors and made some edits. (See also, Lines 236).

L220-221: Did the authors assess or measure the bioelectric potential of the mat used?

Response: As reviewer’s comment, we measured the bioelectric potential of the earthing mattress.

Left panel showed that the potential rises to 229V before using the mat. However, right panel showed that the potential immediately carries a voltage of 0V. It was confirmed that the potential difference formed in the human body when human used the earthing mat.  

Round 2

Reviewer 2 Report

The authors correctly answered the questions asked.

Perhaps, it would have benefited if the authors could show in figure 2 an image of the cages and conditions of the Normal group rats.

Author Response

Reviewer 2 Comments and Suggestions

The authors correctly answered the questions asked. Perhaps, it would have benefited if the authors could show in figure 2 an image of the cages and conditions of the Normal group rats.

  • Response to Reviewer 2:

Thank you for your comment. As Reviewer suggested, we included an image of cages and the conditions of the Normal group rats in the revised version of the manuscript (See also, Figure 2).
